# A General Temperature-Dependent Stress–Strain Constitutive Model for Polymer-Bonded Composite Materials

**DOI:** 10.3390/polym13091393

**Published:** 2021-04-25

**Authors:** Xiaochang Duan, Hongwei Yuan, Wei Tang, Jingjing He, Xuefei Guan

**Affiliations:** 1Graduate School of China Academy of Engineering Physics, 10 Xibeiwang E. Rd., Beijing 100193, China; duanxiaochang19@gscaep.ac.cn; 2Institute of Chemical Materials, China Academy of Engineering Physics, 64 Mianshan Rd., Mianyang 621900, China; yuanhw@caep.cn (H.Y.); tangwei@caep.cn (W.T.); 3School of Reliability and Systems Engineering, Beihang University, 37 Xueyuan Rd., Beijing 100191, China; hejingjing@buaa.edu.cn

**Keywords:** polymer-bonded composites material, stress–strain constitutive, temperature-dependent, nonlinear strain behavior

## Abstract

This study develops a general temperature-dependent stress–strain constitutive model for polymer-bonded composite materials, allowing for the prediction of deformation behaviors under tension and compression in the testing temperature range. Laboratory testing of the material specimens in uniaxial tension and compression at multiple temperatures ranging from −40 ${}^{\circ }$C to 75 ${}^{\circ }$C is performed. The testing data reveal that the stress–strain response can be divided into two general regimes, namely, a short elastic part followed by the plastic part; therefore, the Ramberg–Osgood relationship is proposed to build the stress–strain constitutive model at a single temperature. By correlating the model parameters with the corresponding temperature using a response surface, a general temperature-dependent stress–strain constitutive model is established. The effectiveness and accuracy of the proposed model are validated using several independent sets of testing data and third-party data. The performance of the proposed model is compared with an existing reference model. The validation and comparison results show that the proposed model has a lower number of parameters and yields smaller relative errors. The proposed constitutive model is further implemented as a user material routine in a finite element package. A simple structural example using the developed user material is presented and its accuracy is verified.

## 1. Introduction

Polymer-bonded composites materials (PBMs) have drawn increasing attention in recent years due to their advantage in complex shaping, bi-injection molding, and ease of machining. By compounding different filler materials with polymer, new functional materials can be made, providing a viable technology for on-demand material design and engineering [1]. Applications of PBMs include, but are not limited to, polymer-bonded magnetics [2,3], energetics [4,5], aluminum [6], fiber [7,8], and many others [9,10,11]. To utilize those functional materials in critical applications, understanding their mechanical properties and deformation behaviors under different loading and temperature conditions is of great importance in meeting the basic design criteria.

The mechanical properties such as compliance, modulus, toughness, and strength of polymer materials depend largely on strain (rate) and temperature [12,13]. The stress–strain behavior of PBMs is usually that of viscoelastic plastic, which can be approximated as an elastic (linear) regime followed by a plastic (nonlinear) regime [14] in phenomenological view. The deformation is homogeneous in the elastic regime and can be related to the applying stress by Hooke’s law [15]. Once the stress of the material reaches a certain stress level, the material will produce irreversible strain and plastic deformation [16]. The maximum stress below which the resulting strain remains proportional to the applying stress is the elastic limit of the polymer materials. However, the filler materials can drastically alter the elastic limit and change the stress–strain behavior in both elastic and plastic regimes [17,18,19]. In addition, the plastic regime of the PBM can also be vastly different to that of the regular polymer materials due to the use of different filler materials.

The need to incorporate temperature effects into the deformation behavior of PBMs has been identified in several experimental studies on a variety of materials, for example, EDC37 in the temperature range of −65 ${}^{\circ }$C to 60 ${}^{\circ }$C by Williamson et al. [20], Rowanex between −60 ${}^{\circ }$C and 60 ${}^{\circ }$C by Walley et al. [21], HTPB at low temperatures by Chen et al. [22], short carbon-fiber-reinforced polyether-ether-ketone (SCFR/PEEK) composites between 20 ${}^{\circ }$C and 235 ${}^{\circ }$C by Zheng et al. [23], epoxy mortar at temperature between 80 ${}^{\circ }$C to 210 ${}^{\circ }$C by Vecchio et al. [24], nanoparticle/epoxy nanocomposites between 23 ${}^{\circ }$C to 53 ${}^{\circ }$C by Unger et al. [25], and many others [26,27]. Several models have been reported to describe the mechanical responses of PBM under different temperatures, including the elastic–plastic fracture-mechanism-based model using the glass-transition temperature ($T_{g}$) as a piecewise knot point with a total number of 39 parameters [28], temperature-dependent mesoscale interface-based model [29], and the temperature-dependent visco-hyperelastic model [30]. Models based on the concept of temperature correction factor and its variants have been seen, for example, in [31,32]. Experimental evidence indicates that the mechanical behaviors of PBMs under tension and compression are quite different, and the stress–strain behavior under tension and compression may be described separately [20,21,28]. The aforementioned models in [29,30,31] were established for a single mode, and their applicability to the other mode is unknown. Other models such as the one reported in [28] and its variants requires a dozen parameters, demanding much more testing data for parameter calibration, which may not be realistic in all cases.

The goal of this study is to develop a general temperature-dependent stress–strain constitutive model for PBMs suitable for both tension and compression loading modes with a minimal set of parameters. Compression and tension testing in the temperature range from −40 ${}^{\circ }$C to 75 ${}^{\circ }$C is performed to obtain stress–strain response data. Based on the observation of the stress–strain curves at individual temperatures, a nonlinear model based on the Ramberg–Osgood (RO) relationship is proposed to describe the stress–strain under a given temperature. By correlating the model parameter with temperature, a general temperature-dependent constitutive model can be formulated. The reminder of the study is organized as follows. First, the experimental work is presented in detail and the data in compression and tension are acquired. Next, the constitutive model is developed based on the testing data. Following that, the model validation using both an independent set of testing data and third-party data is performed. The effectiveness of the model is demonstrated, and the performance of the model is compared with a reference model having a comparable number of parameters. The developed constitutive model is further implemented as a user material in a finite element package for structural applications. Finally, conclusions are drawn based on the current results.

## 2. Experimental Testing

Figure 1 presents the overall process of experimental testing, model development, validation, and comparison. In the experiment, test specimens were prepared according to the national standard [33] and were arbitrarily divided into two groups, e.g., 80% for model development and 20% for validation. The testing is performed at various temperatures in the range of −40 ${}^{\circ }$C to 75 ${}^{\circ }$C under tension and compression. In the model development part, a model based on the RO relationship is proposed to describe the stress–strain data under each of the temperatures. The temperature dependence on model parameters is established using the response surface model and the final temperature-dependent constitutive model is formulated. In model validation, the validation data (20%) and third party data are used to verify the effectiveness of the proposed model. In model comparison, the performance of the proposed model is compared with an existing reference model having a comparable number of parameters.

### 2.1. Specimens Preparation and Test Conditions

The composition of the PBM used in this study is barium sulfate of 94 wt.% embedded in 6 wt.% fluororubber binder. The conversion volume percentages are 87.6% and 12.4%, respectively. The size of the filler particles is in the range of 0.5 mm and 3 mm. The test specimens were obtained from a vendor and the actual image of the PBM sample is shown in Figure 2a. The PBM was inspected using cone-beam computed tomography (CT), as shown in Figure 2b, to ensure the integrity before testing. In the shown CT image, taken from a piece of the material sample, the light-colored particles are filler materials and the dark-colored portion are the binder materials. After CT inspection, the cylinder specimens for compression (Φ20 mm × 20 mm) and dog-bone specimens for tension (Φ15 mm × 50 mm) were fabricated according to the standard [33], with the dimensions shown in Figure 3a,b, respectively.

A uniform scheme of design of experiments was used to determine the design points of the temperature. In this study, 20 test conditions are designed in the temperature range of −40 ${}^{\circ }$C to 75 ${}^{\circ }$C and the temperature interval is less than 10 ${}^{\circ }$C, totaling a number of 191 test specimens. The temperatures and the number of specimens at the given temperature are summarized in Table 1. Moreover, one validation specimen is selected for every 4 points, excluding the boundary temperature points.

### 2.2. Test Results

Uniaxial tension and compression tests were performed on a universal testing machine in an environmental chamber at the specified temperature. The loading rate is 0.5 mm/min, representing a quasi-static condition. Figure 4 and Figure 5 present the average of the tensile and compressive stress–strain curves at different temperatures from initial state to rupture. It should be noted that the stress and strain under tension and compression are expressed by positive numbers in this paper.

It is observed from Figure 4 that the ultimate strength decreases significantly as the temperature increases, e.g., from 11.55 MPa at −40 ${}^{\circ }$C to 2.69 MPa at 75 ${}^{\circ }$C. The compression results shown in Figure 5 can be divided into two regimes. As the strain increases, the stress increases monotonically until the ultimate strength is reached, following by the postultimate stage before fracture. With the further increase of the strain, the stress decreases gradually. For compression results, the ultimate strength decreases from 42.41 MPa at −40 ${}^{\circ }$C to 12.64 MPa at 75 ${}^{\circ }$C. The maximum resistance of a material to external forces is called the stress at break [34]. The ultimate strength in tension and compression indicates the failure. Therefore, the data ranging from the initial state to the stress at break are used for the model development. The main transition temperature for the used binder material is $T_{g}$ = 80 ${}^{\circ }$C, which is consistent with that of a similar material [35]. The stress–strain behavior at a higher temperature that is close to $T_{g}$ may be influenced by this transition temperature. This can be observed from Figure 4 and Figure 5, where at higher temperatures, the stress–strain curves start to deviate much more than those at lower temperatures. As the proposed constitutive model is empirical in nature, the developed model does not deal with the mechanism related to the transition temperature.

## 3. Constitutive Model Development

The stress–strain model under single temperature is proposed first, and the temperature dependence of the model parameters is established using the tension and compression data. After that, the general temperature-dependent stress–strain constitutive model is obtained.

### 3.1. Stress–Strain Model under Single Temperature

The stress–strain curves presented in Figure 4 and Figure 5 indicate that the total response consists of a short linear part and a nonlinear part. Therefore, the total strain ε can be written as the linear accumulation of the two parts such as
(1)$$\varepsilon =\varepsilon_{e}+\varepsilon_{p},$$
where $\varepsilon_{e}$ is linear (elastic) strain and $\varepsilon_{p}$ is nonlinear (plastic) strain. For the linear strain, it can be described by the stress σ and Young’s modulus *E* as (σ/E). For the nonlinear part, a general exponential form as in the Ramberg–Osgood relationship [36] can be used. Substituting the linear and nonlinear parts, for monotonic loading,
(2)$$\varepsilon =\varepsilon_{e}+\varepsilon_{p}=\frac{\sigma }{E}+K{\left(\frac{\sigma }{E}\right)}^{n},$$
to determine the adequacy of Equation (Equation 2) in describing the testing data at a single temperature. The testing data of *T* = −40 ${}^{\circ }$C are arbitrarily chosen to fit the model parameters, and the fitted model parameters are presented in Table 2 for both tension and compression cases. Using the fitting parameters, the results of the mean fit and the testing data are presented in Figure 6 for comparison purposes. It can be observed that the model can capture the linear and nonlinear portions of the testing data fairly well in the given strain range for both tension and compression cases, indicating the effectiveness of Equation (Equation 2) for correlating the stress–strain behavior of the PBM under investigation.

### 3.2. Tension

#### 3.2.1. Model Parameters under Different Temperatures

The nonlinear least squares estimator is used to obtain the parameters (E,K,n) in Equation (Equation 2) for tension testing data of each of the specimens at each of the temperatures. For convenience, the parameter lnK instead of *K* is used. The resulting parameters vs. temperatures are presented in the Figure 7. Every solid dot in Figure 7 is the fitted parameter associated with one specimen. From the results of Figure 7, it can be found that the variation of each of the three parameters with temperatures can loosely be divided into two linear regimes. For parameters lnK and *n*, the change points of the two regimes are about −20 ${}^{\circ }$C, indicated by the vertical dash lines in Figure 7a,b. The two parameters decrease linearly from −40 ${}^{\circ }$C to the change point, and increase linearly from that point to the end. For parameter *E*, the change point is also −20 ${}^{\circ }$C but the variation in the two regimes is reversed compared to that of the other two parameters. The scattering in the fitted parameters shown in Figure 7 is caused by the inherent uncertainty in material properties and stochastic nature of the filler particle distribution.

#### 3.2.2. Temperature Dependence Modeling

Based on the above discussion, it can be observed that the temperature dependence can be described using the piecewise linear relationship; therefore, a piecewise linear model is used to correlate the variation of the three parameters with temperatures.
(3)$$\left\{ \;\begin{array}{c} \ln K=\alpha_{1}+\alpha_{2}\cdot T+\alpha_{3}\cdot (T-T_{k1})\cdot H(T-T_{k1})\\ n=\beta_{1}+\beta_{2}\cdot T+\beta_{3}\cdot (T-T_{k2})\cdot H(T-T_{k2})\\ E=\gamma_{1}+\gamma_{2}\cdot T+\gamma_{3}\cdot (T-T_{k3})\cdot H(T-T_{k3}) \end{array},\right.$$
where $\alpha_{i}$, $\beta_{i}$, and $\gamma_{i}$ (i=1,…,3) are fitting coefficients without unit; *T* is the temperature in degrees centigrade; and $T_{ki}$ (i=1,…,3) are the knot points of the piecewise linear function. The term H(x) is the Heaviside step function defined as
(4)$$H\left(x\right)=\{\;\begin{array}{cc} 1 & x\geq 0\\ 0 & \mathrm{otherwise} \end{array}.$$

According to Figure 7, $T_{ki}$(i=1,…,3) are determined as −20 ${}^{\circ }$C. Using the parameter-fitting results presented in Figure 7 and Equation (Equation 3), the temperature dependence fitting coefficients α, β, and γ are obtained using the ordinary least square estimator as
(5)$$\mathrm{Tension}:\;\;\{\;\begin{array}{ccc} \alpha & =[\alpha_{1},\alpha_{2},\alpha_{3}] & =[3.522,-0.06351,0.3691]\\ \beta & =[\beta_{1},\beta_{2},\beta_{3}] & =[1.516,-0.01018,0.05640]\\ \gamma & =[\gamma_{1},\gamma_{2},\gamma_{3}] & =[13,374,70.93,-166.2] \end{array}.$$

Figure 8 presents the temperature dependence fitting results, where the mean fit (solid lines) is shown. It can be seen that the mean fit of the piecewise linear model in Equation (Equation 3) can roughly describe the temperature dependence of the model parameters lnK, *n*, and *E*.

Substituting Equation (Equation 3) into Equation (Equation 2) with parameters in Equation (Equation 5), the final temperature-dependent stress–strain constitutive model under tension is obtained as
(6)$$\mathrm{Tension}:\;\varepsilon =\frac{\sigma }{E(T)}+K\left(T\right){\;[\frac{\sigma }{E(T)}]}^{n(T)},$$
where the temperature terms are
(7)$$\mathrm{Tension}:\;\{\begin{array}{cc} \ln K(T) & =3.522-0.06351\cdot T+0.3691\cdot (T+20)\cdot H(T+20)\\ n(T) & =1.516-0.01018\cdot T+0.05640\cdot (T+20)\cdot H(T+20)\\ E(T) & =13,374+70.93\cdot T-166.2\cdot (T+20)\cdot H(T+20) \end{array}.$$

The temperature-dependent stress–strain model (Equation (Equation 6)) is used to predict the strain given the temperature and stress. The overall residuals on all the tension data are evaluated and presented in Figure 9. The value corresponding to the sum of squared errors (SSE) is $3.0686\times 10^{-4}$ using Equation (Equation 8). The SSE is given as
(8)$$\mathrm{SSE}=\sum_{i=1}^{N}{\left(y_{i}-{\hat{y}}_{i}\right)}^{2},$$
where $y_{i}$ is the actual value, ${\hat{y}}_{i}$ is the model prediction, and i=1,…,N represents the index of a total number of *N* data points. The standard deviation of the residuals is estimated as $5.133\times 10^{-5}$.

### 3.3. Compression

#### 3.3.1. Model Parameters under Different Temperatures

The same fitting and processing method in Section 3.2.1 is applied to the compression data, and the resulting parameters in Equation (Equation 2) under different temperatures are presented in Figure 10.

Similar to the tension results, the variation of parameters with the temperatures under compression are also piecewise linear. Both lnK and *n* are linear proportional to the temperature in the range of −40–40 ${}^{\circ }$C. In the range of 40–75 ${}^{\circ }$C, the two parameters are inversely proportional to the temperature. The parameter *E* shows the same behavior but with a different change point of −5 ${}^{\circ }$C. Therefore, for compression data, the same equation, Equation (Equation 3), is used with $T_{k1}$ = $T_{k2}$ = 40 ${}^{\circ }$C and $T_{k3}$ = −5 ${}^{\circ }$C.

#### 3.3.2. Temperature Dependence Modeling

Using the parameter-fitting results presented in Figure 10 and Equation (Equation 3), the temperature dependence fitting coefficients α, β, and γ are obtained using the ordinary least squares estimator as
(9)$$\mathrm{Compression}:\{\;\begin{array}{ccc} \alpha & =[\alpha_{1},\alpha_{2},\alpha_{3}] & =[13.74,0.1125,-0.4976]\\ \beta & =[\beta_{1},\beta_{2},\beta_{3}] & =[6.797,0.04343,-0.1937]\\ \gamma & =[\gamma_{1},\gamma_{2},\gamma_{3}] & =[7665,0.7296,-69.69] \end{array}.$$

Substituting Equation (Equation 3) into Equation (Equation 2) with parameters in Equation (Equation 9), the final temperature-dependent stress–strain constitutive model under compression is obtained as
(10)$$\mathrm{Compression}:\;\varepsilon =\frac{\sigma }{E(T)}+K\left(T\right)\;{\left[\frac{\sigma }{E(T)}\right]}^{n(T)},$$
where the temperature terms are
(11)$$\mathrm{Compression}:\;\{\begin{array}{cc} \ln K(T) & =13.74+0.1125\cdot T-0.4976\cdot (T-40)\cdot H(T-40)\\ n(T) & =6.797+0.04343\cdot T-0.1937\cdot (T-40)\cdot H(T-40)\\ E(T) & =7665+0.7269\cdot T-69.69\cdot (T+5)\cdot H(T+5) \end{array}.$$

Figure 11 presents the temperature dependence fitting results, where the mean fits (solid lines) are shown. It can be observed that the temperature dependence can be described using the piecewise linear functions. Figure 12 presents the residuals of the model fitting for compression data. The SSE in this case is 0.07323 and the standard deviation of the residuals is 0.001089.

## 4. Model Validations and Comparisons

Independent sets of testing data are used to validate the performance of the proposed model. Third-party data published for a PBM are further used to validate the generality of the model. To compare the proposed model with existing models, a model with a comparable number of parameters is adopted and results from the two models are compared. In the validation and comparison, the following two general measures, namely, the mean absolute error (MAE) and the root mean square error (RMSE), are used.
(12)$$\mathrm{MAE}=\frac{\sum_{i=1}^{n}\left|y_{i}-{\hat{y}}_{i}\right|}{N},$$
(13)$$\mathrm{RMSE}=\sqrt{\frac{\mathrm{SSE}}{N}},$$
where $y_{i}$, ${\hat{y}}_{i}$, *i*, and *N* are defined as in Equation (Equation 8).

### 4.1. Validation Using Independent Sets of Testing Data

The data from the validation specimens (labeled as ‘V’ in Table 1) are not used for model development and are used for validation. It is noted that both tension validation specimens and compression validation specimens are available. We utilize data associated with the validation specimens. Prediction results of the developed model and the actual testing values are compared. In addition, 2 groups of modeling data near the piecewise point are added to compare with the model fitting results. Figure 13 presents the model prediction results for tension specimens compared with testing results. Figure 14 shows the model prediction results for compression specimens compared with testing results. In both modes, a close agreement between the prediction and testing data can be observed.

The maximum MAE and RMSE are $6.200\times 10^{-5}$ and $8.625\times 10^{-5}$ at 25 ${}^{\circ }$C, respectively, for tension data. For compression data, the maximum MAE and RMSE are 0.001741 and 0.002460 at 10 ${}^{\circ }$C, respectively. The prediction performance for each of the temperatures are detailed in Table 3.

### 4.2. Validation Using Third-Party Data

To further verify the generality of the proposed model, third-party testing data on PBX 9502 from [28] consisted of 5 sets of tension data and 5 sets of compression data (at temperatures −52 ${}^{\circ }$C, −20 ${}^{\circ }$C, 20 ${}^{\circ }$C, 50 ${}^{\circ }$C, and 72 ${}^{\circ }$C) are used to validate the generality of the proposed model.

It should be noted that the results of model parameters in Equations (Equation 7) and (Equation 11) cannot be directly used for prediction purposes as those parameters are material specific. The reported data on PBX 9502 are used to fit the developed model using Equation (Equation 2) for tension and compression data at each of the temperatures, and the temperature dependence for both modes are obtained using Equation (Equation 3). The same least squares fitting scheme is used. The prediction results of the proposed model and the actual testing data at the five temperatures are compared and shown in Figure 15. Under tension, the maximum MAE and RMSE are $5.4472\times 10^{-5}$ and $7.8292\times 10^{-5}$ at 50 ${}^{\circ }$C, respectively. For compression, the maximum MAE is 0.001040 at 74 ${}^{\circ }$C and the maximum RMSE is 0.001184 at 20 ${}^{\circ }$C. For both cases, the model can capture the behaviors of the stress–strain variations fairly well.

### 4.3. Model Comparisons

To further investigate the performance of the proposed model, a stress–strain constitutive model reported by Chang et al. [37] is adopted for comparison purposes. This model is formed by appending a quadratic term and a cubic term of the strain to the regular linear model, as in Equation (Equation 14):(14)$$\sigma =E_{0}\varepsilon +C_{1}\varepsilon^{2}+C_{2}\varepsilon^{3},$$
where the parameters $E_{0}$, $C_{1}$, and $C_{2}$ are fitting parameters. It is worth mentioning that other reported models can also be adopted for comparisons; however, this model is chosen due to the fact that the model has a comparative number of parameters and it requires only stress–strain data for fitting. Other models reviewed in the introduction either contain too many parameters or require special testing data.

#### 4.3.1. Tension

The same amount of tension data is used to fit model parameters using Equation (Equation 14) for each of the temperatures. The fitted model parameters $\left(E_{0},C_{1},C_{2}\right)$ are further correlated with temperatures to obtain the following temperature dependence equations in Equation (Equation 15):(15)$$\left\{ \;\begin{array}{cc} E_{0} & =9691-56.78\cdot T-0.9012\cdot T^{2}+0.01122\cdot T^{3}\\ C_{1} & =-2.311\times 10^{6}+74,213\cdot T-878.7\cdot T^{2}-32.35\cdot T^{3}+0.4033\cdot T^{4}\\ C_{2} & =1.061\times 10^{8}-3.471\times 10^{7}\cdot T+5.458\times 10^{5}\cdot T^{2}+16,787\cdot T^{3}-222.5\cdot T^{4} \end{array}.\right.$$

The two models with established model parameters are used to predict the validation data (those labeled as ‘V’ in Table 1), and the results are shown in Figure 16. In general, both models can describe the data satisfactorily; however, for data at 50 ${}^{\circ }$C, the proposed model outperforms the reference model. The errors in terms of MAE and RMSE of the two models are presented in Table 4. From Table 4, it can be seen that the proposed model yields smaller errors for all data sets except the one associated with 0 ${}^{\circ }$C.

#### 4.3.2. Compression

The same amount of data under compression are used to fit the reference model, resulting in the following temperature dependence functions in Equation (Equation 4).
(16)$$\left\{ \;\begin{array}{cc} E_{0} & =8335-80.90\cdot T-1.069\cdot T^{2}+0.02098\cdot T^{3}\\ C_{1} & =-7.200\times 10^{5}+9184\cdot T+106.7\cdot T^{2}-3.504\cdot T^{3}\\ \; & +0.01548\cdot T^{4}-1.420\times 10^{-4}\cdot T^{5}\\ C_{2} & =2.201\times 10^{9}-3.700\times 10^{7}\cdot T-5363\cdot T^{2}+172.5\cdot T^{3} \end{array}.\right.$$

The comparisons for compression data between the two models are made following the same procedure described above and are shown in Figure 17, with the error results compared in Table 5. In general, the proposed model can yield improved prediction results, particularly for the tail regions of the stress–strain data. The detailed error results in Table 5 indicate that the proposed model outperforms the reference model at all temperatures except 10 ${}^{\circ }$C. It should be noted that other probabilistic methods such as Bayesian model assessment [38,39,40] can also be employed to compare the performance of the models.

## 5. A User Material Implementation of the Model

To use the proposed temperature-dependent stress–strain model for structural applications, a user material subroutine for the proposed model is developed for finite element analysis. The dimensional of the structural model is the same as the actual specimens. The structural models for the tension and compression specimens are created and meshed using quadratic hexahedron elements with an average size of 2 mm.

For the tension specimen, a constant stress of 2 MPa is applied to the top face with the bottom face fixed in all degrees of freedom, representing a tension test condition. For the compression specimen, a constant stress of −20 MPa is applied on the top face, and the same boundary conditions are used for the bottom face. Figure 18 demonstrates the resulting normal stresses after applying the load at −40 ${}^{\circ }$C, which are shown for the tension and compression specimens in Figure 18a,b, respectively. Under tension, it can be found that the stress on the top face is about 2 MPa, which is the same as the applied load, and the maximum stress occurs in the middle section of the structural part. Under compression, the maximum stress appears at the fixed position. From the top face, the stress gradually decreases along the UX direction. The stress on the top face is the same as the applied load of 20 MPa. For both cases, the resulting stresses are consistent with the actually applied loads.

Figure 19 presents the FEA results using the specimen structural models in Figure 18 with the developed user material routine for temperatures of −40 ${}^{\circ }$C, 20 ${}^{\circ }$C, and 75 ${}^{\circ }$C and compares them with the theoretical results of the proposed temperature-dependent stress–strain model. Both tension and compression results are compared. The maximum relative error (RE) is 0.01239, occurring at −40 ${}^{\circ }$C for the tension case, and the maximum RE is 0.01537 at 75 ${}^{\circ }$C for the compression case, indicating the effectiveness of the developed user material routine for structural applications.

## 6. Conclusions

A general temperature-dependent stress–strain constitutive model for polymer-bonded composite materials is proposed, developed, validated, and compared. Experimental testing data were acquired in the temperature range of −40–75 ${}^{\circ }$C for both tension and compression. The model under a given temperature is proposed based the linear accumulation of an elastic term and a plastic term, resembling the Ramberg–Osgood relationship. The model parameters are obtained by fitting the testing data at each of the temperatures, and the temperature dependence of the model parameters can be established using the response surface method. The proposed model is validated using independent sets of testing data and third-party data, and is further compared with an existing model with a comparable number of parameters. A user material routine of the proposed model is implemented and verified for structural applications. Based on the current results, the following conclusions are drawn.
The proposed method can unify the temperature effect by modeling the temperature dependence of the constitutive model parameters, thus providing an alternative to the existing temperature correction factor-based methods. Furthermore, such a treatment allows a minimal number of fitting parameters compared to the existing models.The proposed method provides a general approach to model the stress–strain behaviors of PBMs. The basic assumption that the stress–strain response is an accumulation of an elastic term and a power-law plastic term is observed in many other materials such as various metals. The effectiveness and generality of the model are validated and demonstrated by two independent preponderate data sets in the temperature range −40–75 ${}^{\circ }$C.

It is worth mentioning that the developed model is a semiempirical data-driven model, and it does not deal with the microscopic mechanism. The dynamic mechanical analysis on the material can be carried out to enhance the understanding of the mechanism, and SEM imaging may be used to reveal the interface interaction between the filler particles and the binder materials. The constitutive model established in this study does not incorporate the strain rate effect. The strain-rate-induced hardening phenomenon has been reported in several studies, e.g., [41,42]. The incorporation of the strain rate effect to the proposed model will be studied in future work. The proposed model considers the PBM as a homogeneous material in macroscopic perspective. The local variations of the material properties due to the microscopic effects of filler materials and polymer binder interface [43,44,45] may be explored in the future. In addition, the effectiveness of the model is verified using the presented experimental data acquired in this study and a third-party published data. Due to the limit on currently accessible data, the verification of the developed model using other PBMs will be investigated in the future. The current modeling adopts a deterministic approach, and a probabilistic approach can also be employed for reliability analysis [46,47,48].

## Figures and Tables

**Figure 1 polymers-13-01393-f001:**
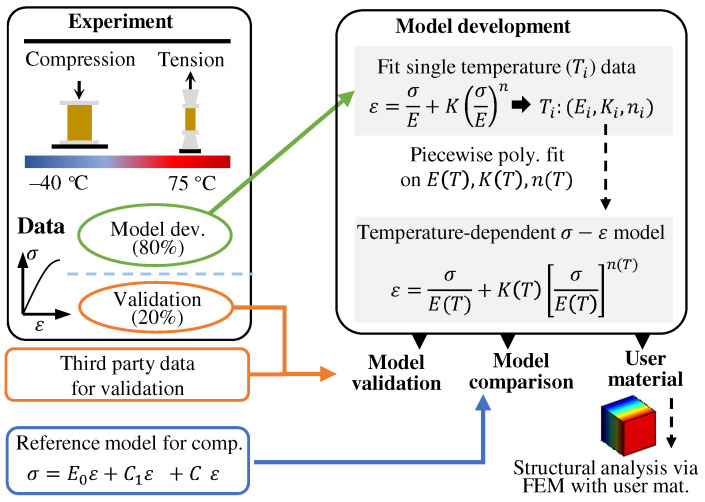
Flow diagram of the study process. There are 5 main parts: experiment, model development, validation, comparison, and FEM usermat.

**Figure 2 polymers-13-01393-f002:**
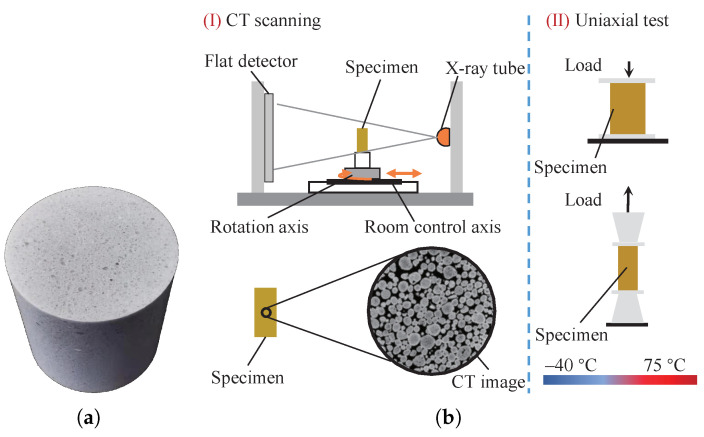
(**a**) A sample of the PBM investigated in this study, and (**b**) schematic diagram of the testing process.

**Figure 3 polymers-13-01393-f003:**
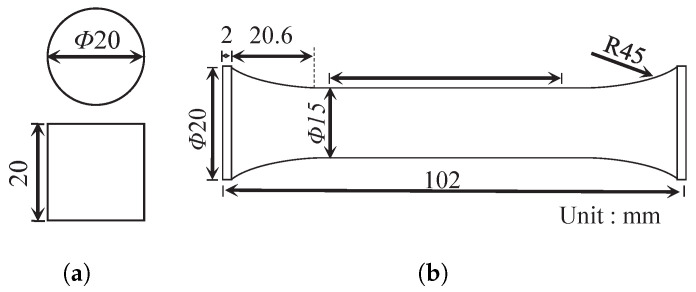
Geometry and dimension of the specimen. (**a**) The compression specimen, and (**b**) the tension specimen.

**Figure 4 polymers-13-01393-f004:**
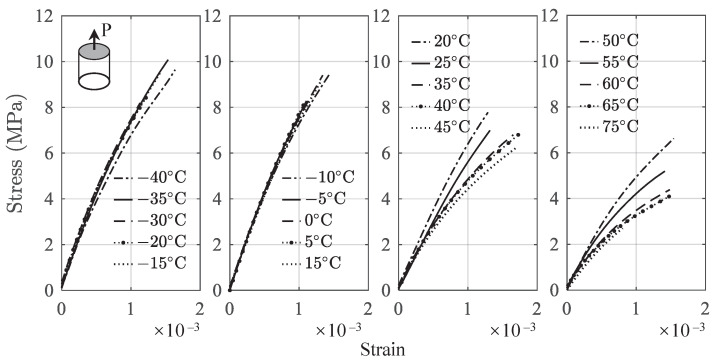
Stress–strain curves under tension.

**Figure 5 polymers-13-01393-f005:**
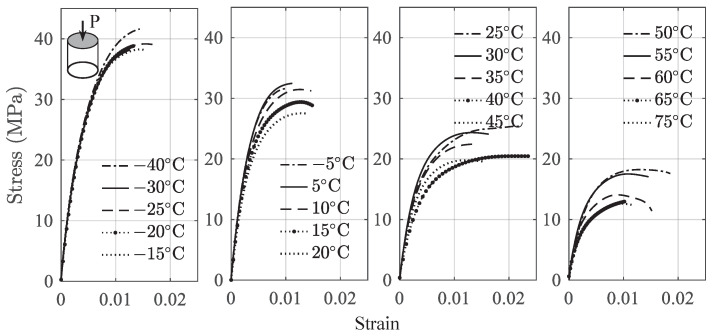
Stress–strain curves under compression.

**Figure 6 polymers-13-01393-f006:**
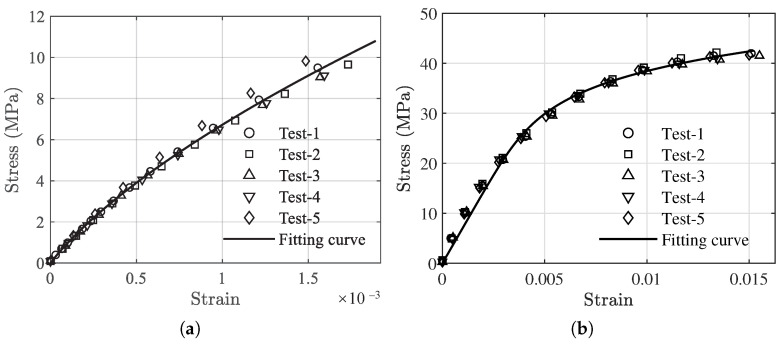
Results of model mean fit using Equation (Equation 2) for testing data at *T* = −40 ${}^{\circ }$C. (**a**) Tension, and (**b**) compression.

**Figure 7 polymers-13-01393-f007:**
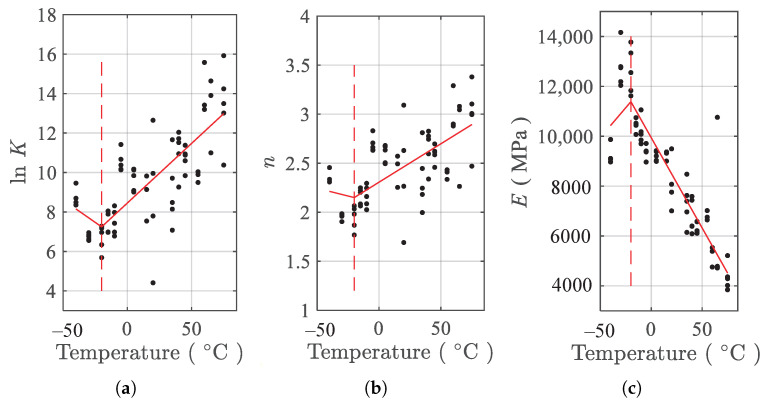
Temperature dependence of parameters under tension. (**a**) lnK, (**b**) *n*, and (**c**) *E*.

**Figure 8 polymers-13-01393-f008:**
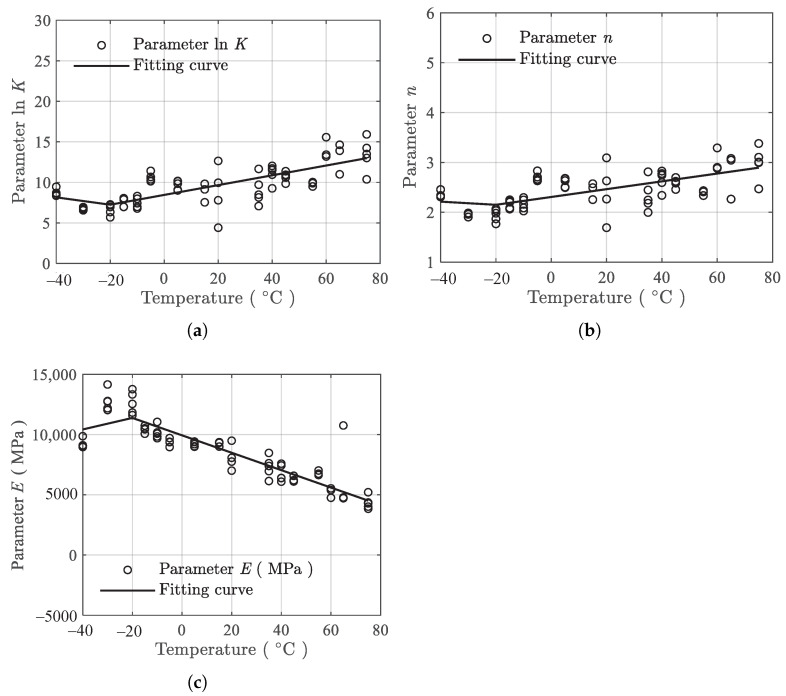
Mean fit of the temperature dependence data under tension. (**a**) lnK, (**b**) *n*, and (**c**) *E*.

**Figure 9 polymers-13-01393-f009:**
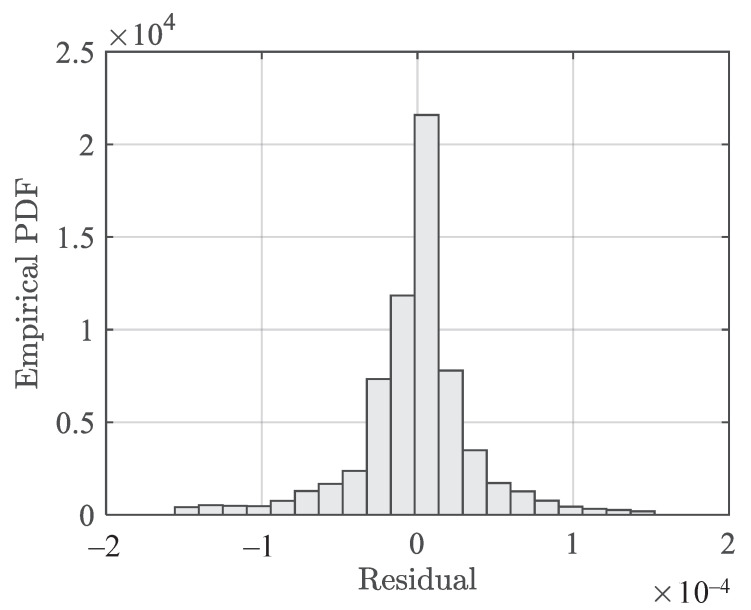
Histogram of residuals of the temperature-dependent stress–strain model (Equation (Equation 6)) on tension data.

**Figure 10 polymers-13-01393-f010:**
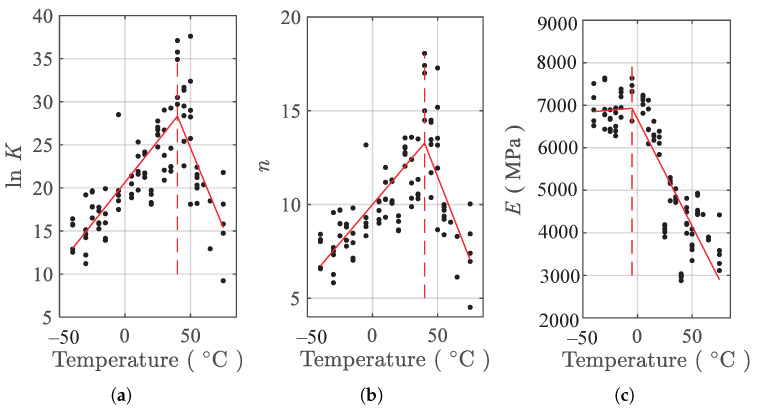
Temperature dependence of parameters under compression. (**a**) lnK, (**b**) *n*, and (**c**) *E*.

**Figure 11 polymers-13-01393-f011:**
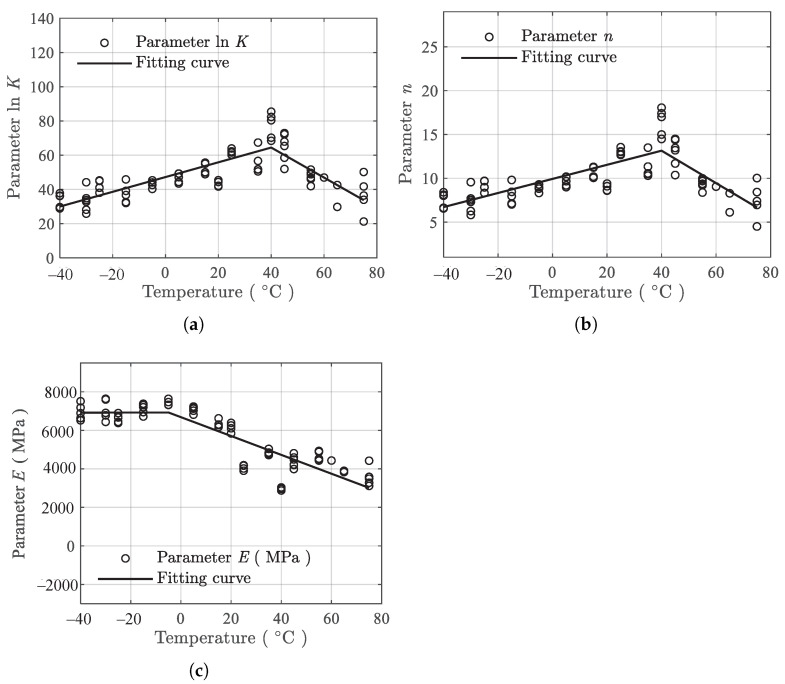
Mean fit of the temperature dependence data under compression. (**a**) lnK, (**b**) *n*, and (**c**) *E*.

**Figure 12 polymers-13-01393-f012:**
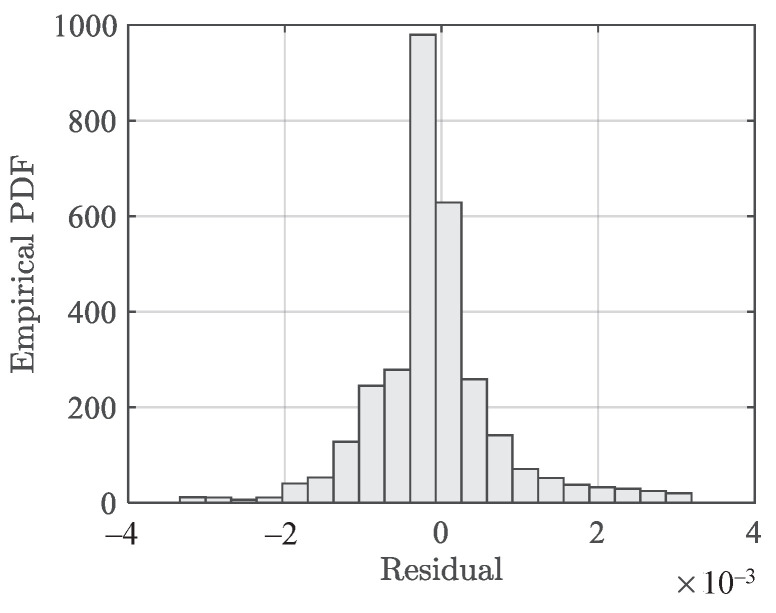
Histogram of residuals of the temperature-dependent stress–strain model (Equation (Equation 10)) on compression data.

**Figure 13 polymers-13-01393-f013:**
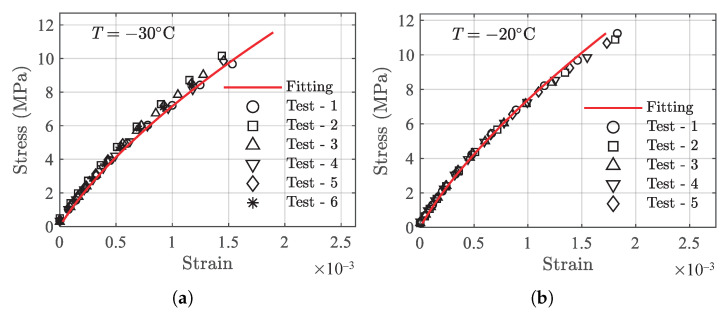
Prediction results of the test data under tension. (**a**,**b**) modeling data; (**c**–**f**) validation data.

**Figure 14 polymers-13-01393-f014:**
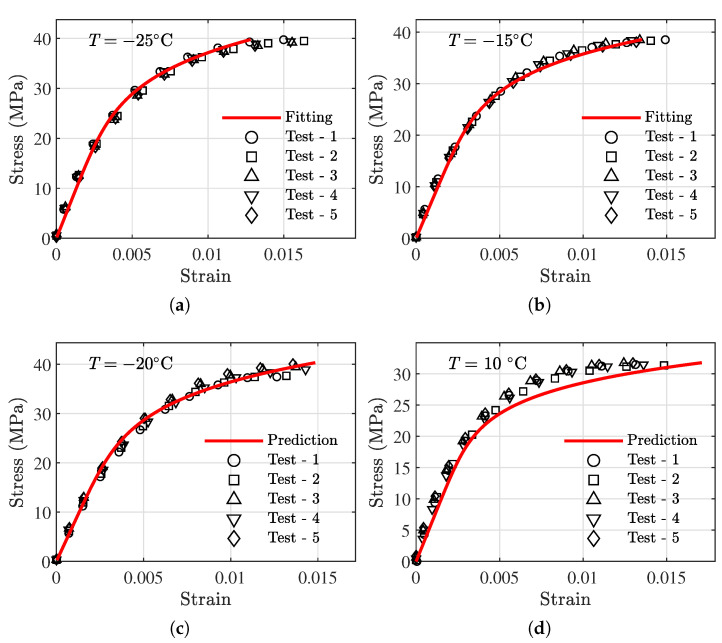
Prediction results of the test data under compression. (**a**,**b**) modeling data; (**c**–**f**) validation data.

**Figure 15 polymers-13-01393-f015:**
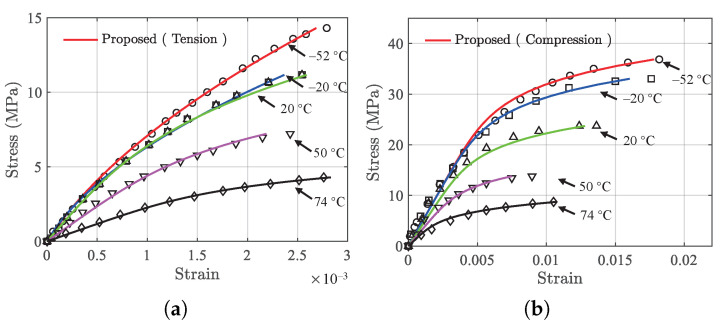
Validation results on third-party data in [28]. Solid lines are the model prediction results and discrete markers are testing data. (**a**) Tension, and (**b**) compression.

**Figure 16 polymers-13-01393-f016:**
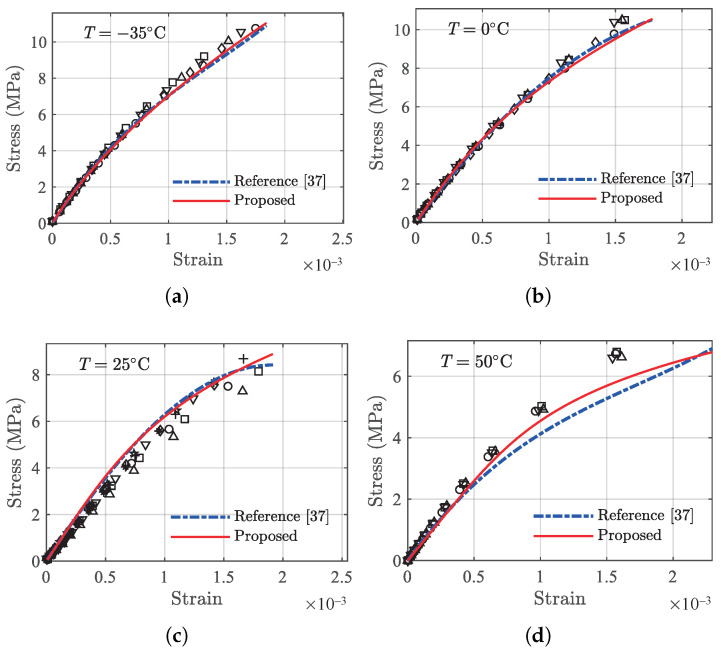
Comparison of the viscoelastic model and the proposed model at validation temperatures under tension. The discrete markers represent testing data. (**a**) −35 ${}^{\circ }$C, (**b**) 0 ${}^{\circ }$C, (**c**) 25 ${}^{\circ }$C, and (**d**) 50 ${}^{\circ }$C.

**Figure 17 polymers-13-01393-f017:**
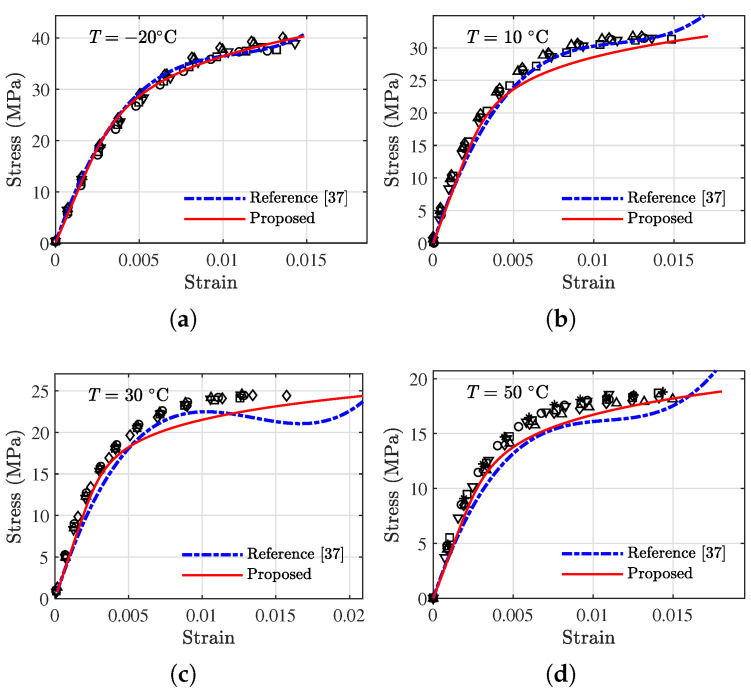
Comparison of the viscoelastic model and the proposed model at validation temperatures under compression. The discrete markers represent testing data. (**a**) −20 ${}^{\circ }$C, (**b**) 10 ${}^{\circ }$C, (**c**) 30 ${}^{\circ }$C, and (**d**) 50 ${}^{\circ }$C.

**Figure 18 polymers-13-01393-f018:**
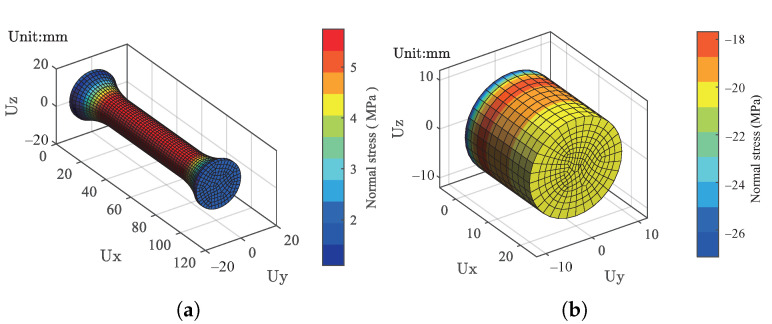
The stress distribution results calculated by the finite element model at −40 ${}^{\circ }$C. (**a**) Tension, and (**b**) compression.

**Figure 19 polymers-13-01393-f019:**
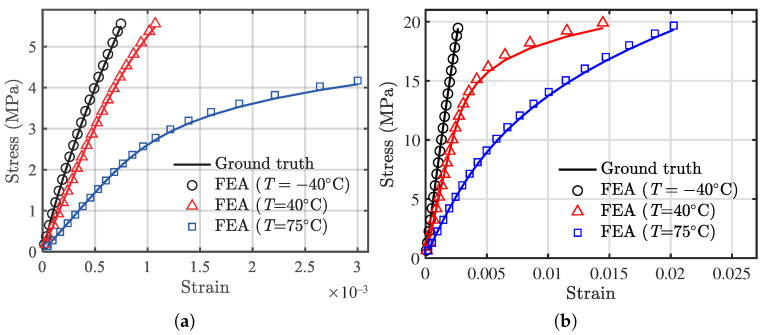
Comparison of FEA results and the proposed model at several temperatures. Solid lines represent ground truth by the proposed model, and discrete markers are numerical results extracted from FEA (user material). (**a**) Tension, and (**b**) compression.

**Table 1 polymers-13-01393-t001:** Test temperature, the number of repeated specimens, and the usage of data. The symbols ‘M’ and ‘V’ denote modeling and validation, respectively. Striking lines denote no testing performed at that temperature.

	Tension		Compression	
**T (${}^{\circ }$C)**	**Num.**	**Usage**	**Num.**	**Usage**
−40	5	M	5	M
−35	5	V		
−30	6	M	6	M
−25			5	M
−20	5	M	5	V
−15	5	M	5	M
−10	5	M		
−5	5	M	5	M
0	5	V		
5	5	M	5	M
10			5	V
15	3	M	5	M
20	4	M	4	M
25	8	V	6	M
30			5	V
35	5	M	5	M
40	5	M	5	M
45	5	M	6	M
50	4	V	6	V
55	3	M	6	M
60	3	M	1	M
65	3	M	2	M
75	5	M	5	M

**Table 2 polymers-13-01393-t002:** Fitting parameters using Equation (Equation 2) and testing data of *T* = −40 ${}^{\circ }$C.

Parameters	Tension	Compression
*E* (MPa)	$1.120\times 10^{4}$	7103
*K*	214.5	$2.664\times 10^{13}$
*n*	1.779	6.955

**Table 3 polymers-13-01393-t003:** Model prediction performance on test data.

Mode	Usage	Temperature (${}^{\circ }$C)	MAE	RMSE
Tension	M	−30	$4.592\times 10^{-5}$	$3.311\times 10^{-5}$
	M	−20	$2.225\times 10^{-5}$	$3.188\times 10^{-5}$
	V	−35	$3.008\times 10^{-5}$	$4.547\times 10^{-5}$
	V	0	$2.139\times 10^{-5}$	$3.328\times 10^{-5}$
	V	25	$6.200\times 10^{-5}$	$8.625\times 10^{-5}$
	V	50	$4.191\times 10^{-5}$	$5.904\times 10^{-5}$
Compression	M	−25	$5.585\times 10^{-4}$	$8.580\times 10^{-4}$
	M	−15	$3.483\times 10^{-4}$	$4.195\times 10^{-4}$
	V	−20	$3.926\times 10^{-4}$	$6.233\times 10^{-4}$
	V	10	0.001741	0.002460
	V	30	0.001100	0.001778
	V	50	0.001195	0.001688

**Table 4 polymers-13-01393-t004:** Comparisons of the performance between the proposed model and the reference model for tension data.

Model	Temperature (${}^{\circ }$C)	MAE	RMSE
Proposed Model	−35	$3.008\times 10^{-5}$	$4.547\times 10^{-5}$
	0	$2.139\times 10^{-5}$	$3.328\times 10^{-5}$
	25	$6.200\times 10^{-5}$	$8.625\times 10^{-5}$
	50	$4.191\times 10^{-5}$	$5.904\times 10^{-5}$
Reference Model [37]	−35	$3.267\times 10^{-5}$	$5.594\times 10^{-5}$
	0	$1.681\times 10^{-5}$	$2.381\times 10^{-5}$
	25	$6.232\times 10^{-5}$	$8.786\times 10^{-5}$
	50	$6.754\times 10^{-5}$	$1.203\times 10^{-4}$

**Table 5 polymers-13-01393-t005:** Comparisons of the performance between the proposed model and reference model for compression data.

Model	Temperature (${}^{\circ }$C)	MAE	RMSE
Proposed Model	−20	$3.926\times 10^{-4}$	$6.233\times 10^{-4}$
	10	0.001741	0.002460
	30	0.001100	0.001778
	50	0.001195	0.001688
Reference Model [37]	−20	$4.722\times 10^{-4}$	$7.916\times 10^{-4}$
	10	$7.109\times 10^{-4}$	$9.527\times 10^{-4}$
	30	0.001262	0.002824
	50	0.002111	0.003094

## Data Availability

All data are included within the text.

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
