# Peer review of "A General Temperature-Dependent Stress–Strain Constitutive Model for Polymer-Bonded Composite Materials"

_polymers, 2021, doi:10.3390/polym13091393_

Round 1

Reviewer 1 Report

The author proposed a temperature–dependent stress–strain constitutive model for polymer–bonded composite materials, to predict their responses under tension and compression loading for a wide range of temperature with a minimal set of parameters. Tests were performed in uniaxial tension and compression loading at multiple temperatures ranging from -40 C to 75 C. Ramberg–Osgood was used a building block to develop the stress–strain constitutive model at single temperature. The efficacy of the proposed model was validated using several independent sets of testing data and literature data. I was concluded that the proposed model yields smaller relative errors and having a smaller number of material parameters.

The paper showed some interesting results on the effect of temperature, but clarity in the tension and compression testing/modeling is needed. Moreover, a very basic phenomenological model was used. The reviewer is hesitant to recommend this for publication in its present form unless a reasonable justification is provided, and the following concerns should be clear.

What filler material was used? Please clearly mention in section 2.1.? More details are needed for the materials. Like filler size distribution etc.

Figure 4 shows the tensile test results, but it looks that the results are overlapping if they plot on the same figure. Can the author justify the reason to plot the results of temperature on the different 4 figures?  Similarly for Figure 5? It was not clear how much the temperature is affecting the results.?

In figure 6 b, the stress at 0.015 strain is reaching more than 40 MPa but in the experimental data figure 5(a) at -40C is well below 40MPa? Why is this contradiction?

The K and n parameters are too much scattered in Figures 7(a) and (b)? Please discuss the fitting of the parameter function. Similarly for compression testing?

In Figure 13: The prediction looks good, but the reviewer wants to see the experimental data at -30 and -25? Apparently, they are very close? Similar to other temperature and compression testing?

In Figure 13, The material parameters were calibrated from -40 to 75?  How the model parameters can be used to predict the performance at -52? Please justify?

Author Response

Reviewer #1: The author proposed a temperature–dependent stress–strain constitutive model for polymer–bonded composite materials, to predict their responses under tension and compression loading for a wide range of temperature with a minimal set of parameters. Tests were performed in uniaxial tension and compression loading at multiple temperatures ranging from -40 C to 75 C. Ramberg–Osgood was used a building block to develop the stress–strain constitutive model at single temperature. The efficacy of the proposed model was validated using several independent sets of testing data and literature data. I was concluded that the proposed model yields smaller relative errors and having a smaller number of material parameters.

The paper showed some interesting results on the effect of temperature, but clarity in the tension and compression testing/modeling is needed. Moreover, a very basic phenomenological model was used. The reviewer is hesitant to recommend this for publication in its present form unless a reasonable justification is provided, and the following concerns should be clear.

RE:

Authors greatly appreciate the reviewer’s comments, and fully understand the reviewer’s concern.

Authors would like to elaborate more on the verification and validation of the model. The justification of the effectiveness of the model in this study was made using both authors’ testing data and third-party data on a different polymer-bonded material.

By splitting the authors’ raw data into the training dataset and validation dataset, the fitting performance was verified using the validation dataset which was not involved in the fitting process. In addition, a third-party dataset reported in Ref. [28] were used to further verify the effectiveness of the model. Results of the two independent sets of data shows the developed model is adequate to describe the stress-strain behavior in the investigated temperature range. The performance of the developed model was further compared with a reported model, and results show that the developed model yields comparable or smaller errors with fewer fitting parameters, compared with the reference model.

Authors fully agree that more validation data on more materials would provide further justification on the model; however, due to the resource limit on acquiring data on other materials, authors plan to investigating the performance of the developed model on other polymer-bonded materials in the future study.

To alleviate the reviewer’s concern, authors have revised the conclusion section by explicitly indicating that the effectiveness of the model is supported by the currently used data, and have added the future work for further verification and validation on other materials. Authors sincerely hope that the reviewer can agree with them.

The above discussion was added in the revised manuscript.

In the last paragraph of Section 6.

“… In addition, the effectiveness of the model is verified using the experimental data presented acquired in this study and a third-party published data. Due to the limit on currently accessible data, the verification of the developed model using other PBMs will be investigated in the future. ...”

Minor concerns,

  1. What filler material was used? Please clearly mention in section 2.1.? More details are needed for the materials. Like filler size distribution etc.

    RE:

    Authors thank the reviewer for the suggestion, and have added the detailed information on the PBM investigated in this study.

    The composition of the PBM used in this study is barium sulfate of 94 wt.% embedded in 6 wt.% fluororubber binder. The size of the filler distribution is in the range of 0.5 mm and 3 mm. The test specimens were obtained from a vendor and the actual image of the PBM sample is shown in Fig. 2(a).

    The above information was added in the revised manuscript in the first paragraph of Section 2.1:

    “… The composition of the PBM used in this study is barium sulfate of 94 wt.% embedded in 6 wt.% fluororubber binder. The conversion volume percentages are 87.6% and 12.4%, respectively. The size of the filler particles is in the range of 0.5 mm and 3 mm. The test specimens were obtained from a vendor and the actual image of the PBM sample is shown in Fig. 2(a). …”

  2. Figure 4 shows the tensile test results, but it looks that the results are overlapping if they plot on the same figure. Can the author justify the reason to plot the results of temperature on the different 4 figures? Similarly for Figure 5? It was not clear how much the temperature is affecting the results.?

    RE:

    Authors thank the reviewer for the comments. Authors have double checked the testing data and the corresponding temperatures to eliminate previous plotting errors.

    To compare the datasets side-by-side, the tension data and the compression data are presented in one plot of Response to Reviewers1.docx, respectively. It is seen that some of the stress-strain curves are close since the step size of the temperature variation in the testing is less than 10℃.

    In the revised manuscript, authors have revised the grouped plots by rescaling the x-axis and y-axis limits to the same ranges in Figures 4-5 for direct visual comparisons. The data are presented as groups with the intention of avoiding congestion and visual clarity.

  3. In figure 6 b, the stress at 0.015 strain is reaching more than 40 MPa but in the experimental data figure 5(a) at -40C is well below 40MPa? Why is this contradiction?

    RE:

    Authors thanks for the comment. The previous figure presented the incorrect data at -40℃. Authors have double checked the data and the corresponding temperature and the error was eliminated in the revised manuscript in Figure 5 and Figure 6(b).

  4. The K and n parameters are too much scattered in Figures 7(a) and (b)? Please discuss the fitting of the parameter function. Similarly for compression testing?

    RE:

    Authors thanks for the suggestion.

    Each of the points  in Figure 7 was obtained using the least square nonlinear fitting with Eq. (2) and one dataset (acquired at one temperature). After that, the fitted parameters are plotted against its temperature to reveal the temperature dependence. The fitting of the temperature dependence relationship is made using the standard linear least square fitting with Eq. (3) and the data points in Figure 7.

    The source of the scatter is the inherent uncertainty in material properties and stochastic distribution of the filler particles. It can be seen that the repeated testing data which conducted at the same temperature also shows noticeable deviations. In addition, a tight y-axis limit in Figure 7 and a high height-to-width ratio of the plots both greatly exaggerate the visual effect of the scattering. In fact, Figure 8 and Figure 7 are the same data but plotted with different y-axis limits.

    The above discussion was added in the revised manuscript:

    In the first paragraph of Section 3.2.1

    “… The scattering in the fitted parameters shown in Fig. 7 is caused by the inherent uncertainty in material properties and stochastic nature of the filler particle distribution….”

    In the first paragraph of Section 3.2.2

    “ … It can be observed that the temperature dependence can be described using the piecewise linear relationship; therefore, a piecewise linear model is used to correlate the variation of the three parameters with temperatures. …”

    In the first paragraph below Eq. (4)

    “… Using the parameter fitting results presented in Fig. 7 and Eq. (3), the temperature dependence fitting coefficients , , and  are obtained using ordinary least square estimator as…”

  5. In Figure 13: The prediction looks good, but the reviewer wants to see the experimental data at -30 and -25? Apparently, they are very close? Similar to other temperature and compression testing?

    RE:

    Authors thank the reviewer for the comment.

    Due to the space limitation required in the initial submitted manuscript those results were omitted.

    Authors have included the results of -30°C, -20°C under tension and -25°C, -15°C under compression in the revised manuscript. See Figure 13(a)-(b), Figure 14(a)-(b) and Table 3 for details.

  6. In Figure 13, The material parameters were calibrated from -40 to 75?  How the model parameters can be used to predict the performance at -52? Please justify?

    RE:

    Authors thank the reviewer for the suggestion. Figure 13 shows the prediction results of validation data of the PBM investigated. The fitted parameters were identified using the authors’ testing data; therefore, the resulting (material-specific) model cannot be directly applied to a different type of material PBX 9502 which associates with the testing dataset acquired at -52 °C.

    The data at -52 °C on PBX 9502 are third-party data published in Ref. [28]. The purpose of Section 4.2 is to demonstrate the developed model can be applied to other PBMs. As the model parameters are material specific, the parameters for predicting PBX 9502 were estimated from the testing data of that material. The prediction for -52 °C is made using the same model of Eq. (2) and temperature dependence of Eq. (3).

    The prediction results of the proposed model and the actual testing data at the five temperatures are compared and shown in Figure 14.

    The above discussion was added in the revised manuscript, in the second paragraph of Section 4.2

    “… It should be noted that the results of model parameters in Eq. (7) and Eq. (11) cannot be directly used for prediction purposes as those parameters are material specific. The reported data on PBX 9502 are used to fit the developed model using Eq. (2) for tension and compression data at each of the temperatures, and the temperature dependence for both modes are obtained using Eq. (3). The same least square fitting scheme is used. The prediction results of the proposed model and the actual testing data at the five temperatures are compared and shown in Fig. 15. …”

    Please see the attachment for more details.

Reviewer 2 Report

This paper deals with a new model able to predict the behavior of polymer-bond composites at various temperatures. This subject is of great interest for researchers and engineers in this field and as such this work is highly relevant, as composites in real-life applications are used seldomly at constant temperature. Nevertheless, some questions arise from the manner and the results presented on this work. I thus recommend major revisions and extensive rework of this paper before acceptance. These revisions are as follows:

1) Please change the polymer/fiber weight percentage to volume percentage as this is the parameter used both for researchers and engineers in the field of composite materials. As such, easier comparisons with available materials can be carried out.

2) Please state the (chemical) nature of the polymer matrix and of the fibers.

3) Concerning the fibers, please state their orientation (UD, woven, random...). This is extremely important as the fiber orientation will have a prominent effect on the mechanical properties, well superior to that of the temperature.

4) Concerning the matrix, please state its secondary and main (glass Tg) transition temperatures. It is well know from Heux, Halary, Leupetre, et.al. that concerning epoxy-based thermosets, the position and the nature of these transitions play a prominent role on the. Indeed, it appears that the proposed model starts to deviate at higher temperatures (as one approaches Tg).

5) Authors need to state the deformation rate applied experimentally. Indeed, modulii, strain and strength are very dependent on the test speed!

6) Dynamic Mechanical Analyses on these materials have to be carried out and are necessary for the understanding of the work. As such, two measurements have to be made:

a) Temperature ramp at 1Hz of frequency from -140°C to 200°C to answer points risen in remark 4)

b) Time-Temperature Superposition Principle measurements centered at the validation temperatures (-35°C, 0°C, 25°C and 50°C) with a comparison of these results at a frequency of 1x10^-4 Hz (comparable to a deformation rate between 5 and 10 mm/min usually applied on such materials).

7) SEM imaging of the samples after testing would be great to better understand the polymer matrix interface interactions and would enrich the paper. Nevertheless they are not crucial for this work.

After these points have been taken into consideration, I would be more than happy to review this paper in a second round.

Author Response

Reviewer #2: This paper deals with a new model able to predict the behavior of polymer-bond composites at various temperatures. This subject is of great interest for researchers and engineers in this field and as such this work is highly relevant, as composites in real-life applications are used seldomly at constant temperature. Nevertheless, some questions arise from the manner and the results presented on this work. I thus recommend major revisions and extensive rework of this paper before acceptance. These revisions are as follows:

Minor concerns,

  1. Please change the polymer/fiber weight percentage to volume percentage as this is the parameter used both for researchers and engineers in the field of composite materials. As such, easier comparisons with available materials can be carried out.

    Authors thanks for the suggestion, and have added the information in the revised manuscript.

    In the first paragraph of Section 2.1:

    “… The composition of the PBM used in this study is barium sulfate of 94 wt.% embedded in 6 wt.% fluororubber binder. The conversion volume percentages are 87.6% and 12.4%, respectively. The size of the filler particles is in the range of 0.5 mm and 3 mm. The test specimens were obtained from a vendor and the actual image of the PBM sample is shown in Fig. 2(a). ...”

  2. Please state the (chemical) nature of the polymer matrix and of the fibers.

    RE:

    Authors thank the reviewer for the suggestion. The composition of the PBM used in this study is barium sulfate of 94 wt.% embedded in 6 wt.% fluororubber binder. The size of the filler particles is in the range of 0.5 mm and 3 mm. No fibers are used in the PBM.

    The above information was added in the revised manuscript.

    In the first paragraph of Section 2.1:

    “… The composition of the PBM used in this study is barium sulfate of 94 wt.% embedded in 6 wt.% fluororubber binder. The conversion volume percentages are 87.6% and 12.4%, respectively. The size of the filler particles is in the range of 0.5 mm and 3 mm. ...”

  3. Concerning the fibers, please state their orientation (UD, woven, random...). This is extremely important as the fiber orientation will have a prominent effect on the mechanical properties, well superior to that of the temperature.

    RE:

    Authors thank the reviewer for such a detailed suggestion. The PBM investigated in this study does not contain fibers. The filler material in the testing specimen in this study is barium sulfate in the form of particles.

  4. Concerning the matrix, please state its secondary and main (glass Tg) transition temperatures. It is well know from Heux, Halary, Leupetre, et.al. that concerning epoxy-based thermosets, the position and the nature of these transitions play a prominent role on the. Indeed, it appears that the proposed model starts to deviate at higher temperatures (as one approaches Tg).

    RE:

    Authors thanks for the comment. The matrix (binder) material of the PBM is mainly consisted of fluororubber. The actual glass transition temperature is around 80°C via testing with a heating rate of 1°C/min. The effect of temperature on the mechanical properties of PBM becomes significant when the testing temperature is close to the glass transition temperature. As the constitutive model is empirical in nature; therefore, the developed model does not deal with the mechanism related to the transition temperature.

    The above discussion was added in the revised manuscript, in the last paragraph of Section 2.2

    The main transition temperature for the used binder material is Tg = 80°C which is consistent with that of a similar material [35]. The stress–strain behavior at a higher temperature that is close to Tg may be influenced by this transition temperature. This can be observed from Figs. 4-5 that at higher temperatures the stress–strain curves start to deviate much more that those at lower temperatures. As the proposed constitutive model is empirical in nature; therefore, the developed model does not deal with the mechanism related to the transition temperature.…

  5. Authors need to state the deformation rate applied experimentally. Indeed, modulii, strain and strength are very dependent on the test speed!

    RE:

    The deformation rate in the testing is 0.5 mm/min. The information is presented in the first paragraph of Section 2.2:

    “… Uniaxial tension and compression tests were performed on a universal testing machine in an environmental chamber at the specified temperature. The loading rate is 0.5 mm/min, representing a quasi–static condition. …”

  6. Dynamic Mechanical Analyses on these materials have to be carried out and are necessary for the understanding of the work. As such, two measurements have to be made:

    1. Temperature ramp at 1Hz of frequency from -140°C to 200°C to answer points risen in remark 4)
    2. Time-Temperature Superposition Principle measurements centered at the validation temperatures (-35°C, 0°C, 25°C and 50°C) with a comparison of these results at a frequency of 1x10^-4 Hz (comparable to a deformation rate between 5 and 10 mm/min usually applied on such materials).

    RE:

    Authors thank the reviewer for the suggestion. Authors agree with the reviewer that dynamic mechanical analyses would enhance the understanding of the stress-strain behavior and revealing the detailed mechanism, and would like to stress the scope and limitation of the proposed model.

    The scope and the focus of the study is to developed a temperature-dependent constitutive model that can be used to describe the stress-strain behavior in the investigated temperature range. Consequently, the model is a semi-empirical data-driven model. In other words, the model can not explain the detailed mechanism from a physical/chemical point of view.

    The model essentially identifies the needed parameters using testing standard stress-strain testing data. Therefore, the suggested testing results, although valuable, cannot be directly incorporated into the model due to the empirical and data-driven nature of the model. Due to the resource limit and accessibility to identical samples, authors cannot perform the suggested testing in a timely manner. More detailed mechanism-oriented study may be carried out when resource becomes available. Authors sincerely hope the reviewer can agree with them.

    The scope and limitation of the model has been re-emphasized in the revised manuscript, in the last paragraph of Section 6.

    It is worth mentioning that the developed model is a semi-empirical data-driven model, and it does not deal with the microscopic mechanism. The dynamic mechanical analysis on the material can be carried out to enhance the understanding of the mechanism, and the SEM imaging may be used to reveal the interface interaction between the filler particles and the binder materials. The constitutive model established in this study does not incorporate the strain rate effect. The strain rate–induced hardening phenomenon has been reported in several studies, e.g., Refs. [41,42]. The incorporation of the strain rate effect to the proposed model will be studied in the future work.  …

  7. SEM imaging of the samples after testing would be great to better understand the polymer matrix interface interactions and would enrich the paper. Nevertheless they are not crucial for this work.

    RE:

    Authors thank the reviewer for the suggestion, and agree with the reviewer that the SEM can provide a means of understanding the filler/binder interface interactions. Due to the limit access to the required testing facilities, authors could not perform more detailed microscopic investigation in the current condition. Therefore, authors have explicitly stressed the empirical nature of the developed model and emphasized that the model cannot handle the mechanism explanation in a microscopic scale. Authors hope the reviewer can agree with them on the limitation and scope of the developed model.

    The above discussion has been emphasized in the revised manuscript in the last paragraph of Section 6.

    “…It is worth mentioning that the developed model is a semi-empirical data-driven model, and it does not deal with the microscopic mechanism. The dynamic mechanical analysis on the material can be carried out to enhance the understanding of the mechanism, and the SEM imaging may be used to reveal the interface interaction between the filler particles and the binder materials. The constitutive model established in this study does not incorporate the strain rate effect. The strain rate–induced hardening phenomenon has been reported in several studies, e.g., Refs. [41,42]. The incorporation of the strain rate effect to the proposed model will be studied in the future work. The proposed model considers the PBM as a homogeneous material in macroscopic perspective. The local variations of the material properties due to microscopic effects of filler materials and polymer binder interface [43–45] may be explored in the future. In addition, the effectiveness of the model is verified using the experimental data presented acquired in this study and a third-party published data. Due to the limit on currently accessible data, the verification of the developed model using other PBMs will be investigated in the future. …”

Reviewer 3 Report

General comments: The goal of this study is to develop a general temperature-dependent stress-strain constitutive model for PBMs suitable for both tension and compression loading modes with a minimal set of parameters.

Specific Issues:  

  1. Please eliminate multiple references. After that, please check the manuscript thoroughly and eliminate ALL the lumps in the manuscript. This should be done by characterising each reference individually and by mentioning 1 or 2 phrases per reference to show how it is different from the others and why it deserves mentioning. Multiple references are of no use for a reader and can substitute even a kind of plagiarism, as sometimes authors are using them without proper studies of all references used. In the case, each reference should be justified by it is used and at least short assessment provided.
  2. In the methodology section add details of FEM modelling. 
  3. What is the basis of the design of  experiment, please elaborate.
  4. Please explain Figure 5 to 8 by considering physical significance. 
  5. Reason of including section 4.2 ?, as validation is already explain by Section 4.1 .
  6. Figure 18: The FEM model must be briefly explain here . Temperature data of -40 degree is presented why? , Other temperature curve was not explained why? 

Author Response

Reviewer #3: The goal of this study is to develop a general temperature-dependent stress-strain constitutive model for PBMs suitable for both tension and compression loading modes with a minimal set of parameters.

Specific Issues,

  1. Please eliminate multiple references. After that, please check the manuscript thoroughly and eliminate ALL the lumps in the manuscript. This should be done by characterising each reference individually and by mentioning 1 or 2 phrases per reference to show how it is different from the others and why it deserves mentioning. Multiple references are of no use for a reader and can substitute even a kind of plagiarism, as sometimes authors are using them without proper studies of all references used. In the case, each reference should be justified by it is used and at least short assessment provided.

    RE:

    Authors thanks for the suggestion. All unnecessary lumps were removed and marked in the revised manuscript.

    The general background information references such as those mentioning various types of PBMs are kept as simple forms as those are only used to support the wide applications and the significance of PBMs. Such references are Refs. [2-11].

    Authors have balanced the general and necessary details in the revised manuscript, regarding on the need for incorporating temperature effect, as the following:

    “ … The need for incorporating temperature effects into the deformation behavior of PBMs have been identified in several experimental studies on a variety of materials, for example, EDC37 in the temperature range of −65◦C to 60◦C by Williamson et al. [20], Rowanex between −60◦C and 60◦C by Walley et al. [21], HTPB at low temperatures by Chen et al. [22], short carbon fiber reinforced polyether-ether-ketone (SCFR/PEEK) composites between 20◦C and 235◦C by Zheng et al. [23], epoxy mortar at temperature between 80◦C to 210◦C by Vecchio et al. [24], nanoparticle/epoxy nanocomposites between 23◦C to 53◦C by Unger et al. [25], and many others [26,27]. Several models have been reported to describe the mechanical responses of PBM under different temperatures, including the elastic–plastic fracture mechanism–based model using the glass–transition temperature (Tg) as a piecewise knot point with a total number of 39 parameters [28], temperature–dependent mesoscale interface–based model [29], and the temperature–dependent visco–hyperelastic model [30]. Models based on the concept of temperature correction factor and its variants have been seen, for example, in Refs. [31,32]. Experimental evidence indicates that the mechanical behaviors of PBMs under tension and compression are quite different, and the stress–strain behavior under tension and compression may be described separately [20,21,28]. The aforementioned models in Refs. [29–31] were established for a single mode, and their applicability to the other mode is unknown. Other models such as the one reported in Ref. [28] and its variants requires a dozen of parameters, demanding much more testing data for parameter calibration which may not be realistic in all cases. … ”

  2. In the methodology section add details of FEM modelling.

    RE:

    Authors thanks for the suggestion, and have included the detailed information on the FEM model in the first paragraph of Section 5:

    “… The dimensional of the structural model is the same as the actual specimens. The structural models for the tension and compression specimens are created and meshed using quadratic hexahedron elements with an average size of 2 mm.

    For the tension specimen, a constant stress of 2 MPa is applied to the top face with the bottom face fixed in all degrees of freedom, representing a tension test condition. For the compression specimen, a constant stress of 20 MPa is applied on the top face, and the same boundary conditions are used for the bottom face....”

  3. What is the basis of the design of experiment, please elaborate.

    RE:

    Authors thanks for the suggestion. There is a uniform experimental design method in this paper. In order to obtain a more accurate model, 20 test conditions are designed in the temperature range of -40°C to 75°C and the temperature interval is less than 10°C, totaling a number of 191 test specimens. Moreover, one validation specimen is selected for every 4 points, excluding the boundary temperature points or piecewise points.

    The above discussion was added in the revised manuscript, in the last paragraph of Section 2.1:

    “… A uniform scheme of design of experiments is used to determine the design points of the temperature. In this study 20 test conditions are designed in the temperature range of -40°C to 75°C and the temperature interval is less than 10°C, totaling a number of 191 test specimens. The temperatures and the number of specimens at the given temperature are summarized in Table 1. Moreover, one validation specimen is selected for every 4 points, excluding the boundary temperature points or piecewise points.”

  4. Please explain Figure 5 to 8 by considering physical significance.

    RE:

    Authors thank the reviewer for the suggestion.

    Figure 5 is stress-strain curves under compression, the compression results can be divided into two regimes. As the strain increases, the stress increases monotonically until the ultimate strength is reached, following by the post–ultimate stage before fracture. With the further increase of the strain, the stress decreases gradually. For compression results, the ultimate strength decreases from 42.41 MPa at -40°C to 12.64 MPa at 75°C.

    The above information was added in the revised manuscript, in the second paragraph of Section 2.2:

    “… As the strain increases, the stress increases monotonically until the ultimate strength is reached, following by the post–ultimate stage before fracture. With the further increase of the strain, the stress decreases gradually. …”

    Figure 6 presents the model mean fit using Eq. (2) for testing data at -40°C. It is used to demonstrate that the model in Eq. (2) can adequately describe the stress-strain behavior of the PBM at single temperature.

    The following description was added in the revised manuscript, in the last paragraph of Section 3.1

    “…To determine the adequacy of Eq. (2) in describing the testing data at single temperature. The testing data of T = −40◦C are arbitrarily chosen to fit the model parameters, and the fitted model parameters are presented in Table 2 for both tension and compression cases. Using the fitting parameters, the results of the mean fit and the testing data are presented in Fig. 6 for comparison purposes. It can be observed that the model can capture the linear and nonlinear portions of the testing data fairly well in the given strain range for both tension and compression cases, indicating the effectiveness of Eq. (2) for correlating the stress–strain behavior of the PBM under investigation. …”

    Figure 7 is the temperature dependence of parameters under tension. It presents the distribution of parameters (E, K, n) at different temperatures. The following discussion was modified to elaborate the results in the revised manuscript, in the first paragraph of Section 3.2.1:

    “… From the results of Fig. 7, it can be found that the variation of each of the three parameters with temperatures can loosely be divided into two linear regimes. For parameters lnK and n, the change points of the two regimes are about −20◦C, indicated by the vertical dash lines in Fig. 7(a) and Fig. 7(b). The two parameters decrease linearly from −40◦C to the change point, and increase linearly from that point to the end. For parameter E, the change point is also −20◦C but the variation in the two regimes is reversed comparing to that of the other two parameters. The scattering in the fitted parameters shown in Fig. 7 is caused by the inherent uncertainty in material properties and stochastic nature of the filler particle distribution. …”

    Figure 8 is the mean fit of the temperature dependence data under tension. It can be observed that the temperature dependence can be described using the piecewise linear functions, and the dependence of parameters on temperature is consistent with that discussed in Section 3.2.1.

    The above information was added in the revised manuscript, in the paragraph below Eq. (5):

    “… Fig. 8 presents the temperature dependence fitting results, where the mean fits (solid lines) are shown. It can be seen that the mean fit of the piecewise linear model in Eq. (3) can roughly describe the temperature dependence of the model parameters lnK, n, and E….”

  5. Reason of including section 4.2 ?, as validation is already explain by Section 4.1.

    RE:

    Authors thank the reviewer for the comment. In Section 4.2, the third-party data are used to describe the stress-strain behavior of another PBM based on the proposed model. The purpose is to demonstrate that the proposed model is general enough to be used for other PBMs.

    The above information has been stressed in the first paragraph of Section 4.2

    “ … To further verify the generality of the proposed model, third–party testing data on PBX 9502 from Ref. [28], consisted of 5 sets of tension data and 5 sets of compression data, (at temperature -52°C, -20°C, 20°C, 50°C and 72°C ) are used to validate the generality of the proposed model.…”

  6. Figure 18: The FEM model must be briefly explain here . Temperature data of -40 degree is presented why? , Other temperature curve was not explained why?

    RE:

    Authors thanks for the suggestion.

    The purpose of Section 5 is to demonstrate that the developed constitutive model can be implemented as a user-material in commercial FEA package to perform structural analysis for a given structural part.

    The following detailed information regarding FEA analysis was added in the revised manuscript, in the first and second paragraphs of Section 5.

    “…To use the proposed temperature–dependent stress–strain model for structural applications, a user material subroutine for the proposed model is developed for finite element analysis. The dimensional of the structural model is the same as the actual specimens. The structural models for the tension and compression specimens are created and meshed using quadratic hexahedron elements with an average size of 2 mm.

    For the tension specimen, a constant stress of 2 MPa is applied to the top face with the bottom face fixed in all degrees of freedom, representing a tension test condition. For the compression specimen, a constant stress of 20 MPa is applied on the top face, and the same boundary conditions are used for the bottom face. Fig. 18 demonstrates the resulting normal stresses after applying the load at -40◦C are shown in for the tension and compression specimens in (a) and (b), respectively. …”

    The reason why only the stress distribution at - 40 ℃ is given is due to the fact that the sole purpose of such a 3D plot is to illustrate the stress distribution in the structural and visually prove the effectiveness of the FEM.

    The detailed comparisons for other temperatures are presented in Figure 19, which provides useful quantitative information. The information conveyed by Figure 9 is more informative than those in Figure 18 for comparison purposes; therefore, the revised manuscript only uses one set of 3D results (in Figure 18) to demonstrate the developed constitutive model is indeed implemented for structural analysis.

    Authors would like to elaborate more details on the structural FEA using the developed constitutive model via user-material element. The following detailed information has been added in the revised manuscript, in the second paragraph of Section 5.

    “… Under tension, it can be found that the stress on the top face is about 2 MPa, which is the same as the applied load, and the maximum stress occurs in the middle section of the structural part. Under compression, the maximum stress appears at the fixed position. From the top face, the stress gradually decreases along the UX direction. The stress on the top face is the same as the applied load of 20 MPa. For both cases, the resulting stresses are consistent with the actually applied loads. ”

Round 2

Reviewer 1 Report

The authors have satisfactorily addressed all the concerns.

Reviewer 2 Report

I accept the manuscript in its revised form,

Kind regards